# First Isolation of Punique Virus from Sand Flies Collected in Northern Algeria

**DOI:** 10.3390/v14081796

**Published:** 2022-08-17

**Authors:** Hemza Manseur, Aissam Hachid, Ahmed Fayez Khardine, Kamal Eddine BENALLAL, Taha Bia, Merbouha Temani, Ahcene HAKEM, Maria Paz Sánchez-Seco, Idir Bitam, Ana Vázquez, Ismail LAFRI

**Affiliations:** 1Institut des Sciences Vétérinaires, Université de Blida 1, Blida 09000, Algeria; 2Laboratoire des Biotechnologies Liées à la Reproduction Animale (LBRA), Institut des Sciences Vétérinaires, Université de Blida 1, Blida 09000, Algeria; 3Laboratoire des Arbovirus et Virus Emergents, Institut Pasteur d’Algérie, Alger 16000, Algeria; 4Faculté de Pharmacie, Univérsité d’Alger 1, Alger 16000, Algeria; 5Laboratoire d’Éco-Épidémiologie Parasitaire et de Génétique des Populations, Institut Pasteur d’Algérie, Alger 16000, Algeria; 6Institut des Sciences Vétérinaires, Université de Tiaret, Tiaret 14000, Algeria; 7Centre de Recherche en Agropastoralisme (CRAPast) Djelfa, Djelfa 17000, Algeria; 8Arbovirus and Imported Viral Diseases Laboratory, National Center of Microbiology, Instituto de Salud Carlos III, 28001 Madrid, Spain; 9CIBER Enfermedades Infecciosas (CIBERINFEC), 28001 Madrid, Spain; 10Ecole Supérieure des Sciences de l’Aliment et des Industries Agroalimentaires, Alger 16000, Algeria; 11CIBER Epidemiología y Salud Pública (CIBERESP), 28001 Madrid, Spain

**Keywords:** Algeria, Kherrata, phlebovirus, *Phlebotomus perniciosus*, Punique virus, sand flies, molecular epidemiology

## Abstract

In the last decade, several phleboviruses transmitted by sand flies were detected in the Mediterranean countries, with the health impact of some of them being unknown. From September to October 2020, a total of 3351 sand flies were captured in Kherrata (Bejaia, northern Algeria) and identified by sex, grouped in 62 pools, which were tested for the presence of phlebovirus RNA using endpoint RT-PCR. Two pools (male and female, respectively) were positive. The genome sequencing and phylogenetic analysis showed that the two phleboviruses detected were closely related to the Punique virus (PUNV) isolated in Tunisia and detected in Algeria. Both PUNV strains were isolated on VERO cells from positive pools. Morphological identification of 300 sand flies randomly selected, showed a clear dominance of *Phlebotomus perniciosus* (98.67%). The dominance of this species in the study area was confirmed by PCR targeting the mitochondrial DNA. Our result represents the first isolation of PUNV and the second report in Algeria from two distinct regions which confirm its large circulation in the country and more broadly in North Africa. Further studies are needed to measure the impact on public health through seroprevalence studies in humans as well as animals and to investigate its potential involvement in neurological viral diseases.

## 1. Introduction

To date, 24 sand flies species belonging to *Phlebotomus* and *Sergentomyia* genera were recorded in Algeria and some species are implicated mainly in the transmission of leishmaniasis and also arboviruses [1,2]. Sand fly-borne phleboviruses (SBPs) are wide world distributed with varying epidemiology and public health threat. They are enveloped, spherical viruses, belonging to the genus *Phlebovirus,* (family *Phenuiviridae*: order Bunyavirales). Their genome is a single-stranded negative-sense RNA composed of three segments designated as large (L), medium (M), and small (S), they encode for the RNA-dependent RNA polymerase (L-segment), the envelope glycoproteins Gn and Gc, and the non-structural protein NSm (M-segment), and the nucleocapsid protein (N) and a smaller non-structural protein (NSs) (S-segment) [3,4,5]. The International Committee on Taxonomy of Viruses (ICTV) currently recognizes 66 species within the genus *Phlebovirus* [6], among them, 42 are exclusively or partially transmitted by females sand flies during the blood meal [7,8]. These viruses are found in different geographic locations, in the new world (Bujaru virus, Candiru virus, Chilibre virus, Frijoles virus, and Punta Toro virus) and sand fly fever Naples virus (SFNV), sand fly fever Sicilian virus (SFSV), Salehabad virus and Punique virus (PUNV) are reported around the Mediterranean basin, central and western Asia [9]. In the Mediterranean area, SFNV and SFSV are recognized for their clinical importance. They are the causative agents of a mild three-day fever in humans, also known as *pappataci* fever [10]. Nevertheless, the Toscana virus (TOSV), classified in the SFNV species, is associated with more severe illnesses such as meningitis or meningoencephalitis during the summer season [11]. In addition, other SBPs were detected in humans; this suggests a large spectrum of SBPs capable to produce human infections [12]. In North Africa, many SBPs have been reported. TOSV was detected and isolated in the Maghreb region except for Mauritania and Libya [13,14,15,16,17]. PUNV, a newly described SBPs species, was isolated for the first time in 2008 from *Ph. Perniciosus* and detected in *Ph. Longicuspis* in the vicinity of Utique, Tunisia [18]. However, in Algeria, few data on SBP_S_ are available, in 2006 Sicilian-like virus was detected for the first time in *Ph. Ariasi* and later in *Ph. Papatasi* [19,20]. Followed by Naples-like virus RNA detected in *Ph. Longicuspis* [20]. In 2013, TOSV was also detected and isolated in Draa El Mizan (Tizi-Ouzou) from one pool of sand flies [17]. Concerning the circulation of these viruses among the populations, only three seroprevalence studies were conducted in Larbaa Nath Iraten (Tizi-Ouzou) and Bousmail (Tipaza), northern Algeria, to detect the circulation of SFNV and SFSV, which revealed that 21.6% and 5% of the population were in contact with these viruses, respectively [19,20]. Recently, Alkan et al. showed that 50% of the tested human sera were positive for TOSV [17]. Regarding animals, the serological survey on dogs revealed that 4.5% of the tested animals were positive for TOSV [21]. Leishmaniasis and phleboviruses are both transmitted by sand flies. Furthermore, *Leishmania infantum* and TOSV share the same vector *Ph. Perniciosus* [22]. Moreover, a co-circulation of phleboviruses and *Leishmania* parasites has been recently demonstrated in humans and sand flies in Tunisia and Italy [23,24]. In this context, the aim of this study is to investigate the circulation of phleboviruses among sand flies in a well-known endemic area of human and canine leishmaniasis [25], to understand better the circulation of SBP_S_ in Algeria.

## 2. Materiel and Methods

### 2.1. Ethical Considerations

Heads of households who were chosen for sampling sand flies from inside their homes and animal shelters provided verbal informed consent.

### 2.2. Sand Flies Sampling

The present study was conducted in Kherrata, (36°24′20″ N, 5°16′37″ E), in northern Algeria (Figure 1), between September and October 2020. Regarding sand flies activity, in northern Algeria, two peaks have been reported annually, the first in July–August and the second in late September/early October [26]. Sand flies were caught using CDC miniature light traps (Centres for Disease Control, John W. Hock Company, Gainesville, FL, USA, CDC), which have been modified with an ultra-fine mesh to make them ideal for sand fly capture [27]. The traps were hung 1 to 2 m above the ground in human habitations and animal shelters. The traps were set at sunset till sunrise the next morning. All sand flies were sorted according to their sex, date of collection, and location, and then, pooled to obtain 20 to 50 sand flies per 1.5 mL tube. All specimens were stored in liquid nitrogen during transport to the laboratory, where they were kept at −80 °C until processed. To reduce handling and to foster virus isolation, no morphological identification of the captured sand flies was carried out before virus testing. All experimentations have been performed at Laboratoire des Arbovirus et Virus Emergents of Institut Pasteur d’Algerie.

### 2.3. Phlebovirus Screening

#### 2.3.1. Sand Flies Processing and Viral RNA Extraction

Each pool was homogenized in 1 mL of Leibovitz’s L15 medium (supplemented with 20% of FBS (fetal bovine serum), 10% of tryptose phosphate, 1% of penicillin/streptomycin, and 0.005% of fungizone) using an MM200 mixer mill (Retsch, Haan, Germany). After clarification, the supernatant was divided into 3 aliquots and stored at −80 °C. The DNA-RNA Extraction MiniKit (Da An Gene Co., Ltd. Of sun Yat University. Guangzhou, China) was used to extract viral nucleic acid from 200 µL sand flies homogenates. The extraction was carried out following the manufacturer’s protocol. The nucleic acids were immediately frozen at −80 °C.

#### 2.3.2. Phlebovirus Detection

A generic Reverse Transcription (RT)-Nested-PCR developed by Sánchez-Seco et al. [28] was used for phleboviruses screening in extracted pools, with some modifications. Reverse transcription and subsequent amplification steps were performed with primers targeting the polymerase gene of the segment L:

Nphlebo1+:5′-ATGGARGGITTTGTIWSICIICC-3′, Nphlebo1-:5′-AARTTRCTIGWI GCYTTIARIGTIGC-3′, for the RT-PCR and (Nphlebo2+:5′-WTICCIAAICCI YMSAARATG-3′), (Nphlebo2-:5′TCYTCYTTRTTYTTRARRTARCC-3′) for the Nested-PCR.

First round RT-PCR was conducted by adding 5 µL of the extracts to 45 µL of a mix containing 2× Mix Buffer, 2.4 mM MgSO_4_, 0.4 mM of each dNTP, and 20 pmol of each primer (Phlebo1+ and Phlebo1−), and 2U of the SuperScript™ III One-Step RT-PCR System with Platinum™ Taq DNA Polymerase (Invitrogen, Carlsbad, CA, USA). The RT-PCR cycling program consisted of 50 °C for 15 min and 95 °C for 2 min, followed by 40 cycles at 94 °C for 30 s, annealing temperature at 45 °C for 1 min, and 68 °C for 45 s, with the final elongation step at 68 °C for 5 min. Second-round Nested-PCR was performed with 1 µL of the RT-PCR product added to 49 µL of a Mix containing 10× Mix Buffer- Mg Cl_2_, 0.2 mM Mg Cl_2_, 0.2 mM of each dNTP, 20 pmol of each primer (Phlebo2+ and Phlebo2-), and 2U of Invitrogen™ Platinum™ Taq DNA Polymerase (Invitrogen, Carlsbad, CA, USA), under the following thermal conditions: denaturation temperature at 95 °C for 2 min, followed by 40 cycles of 94 °C for 30 s annealing temperature at 45 °C for 2 min, and 72 °C for 30 s, finishing with an elongation step at 72 °C for 10 min. PCR products of first and second-round amplification were detected by gel-electrophoresis with a 2% agarose gel in TBE buffer.

### 2.4. Virus Isolation

Sand flies homogenates from phlebovirus PCR-positive pools were seeded on Vero African green monkey kidney cells (Vero ATCC CCL81) to attempt virus isolation. To summarize, 800 µL of Eagle’s minimal essential medium (EMEM without FBS) was used to dilute 200 µL of each homogenate, the resulting dilution was subsequently filtered with a 0.22 µL filter. Then, 500 µL of the dilution was inoculated into Vero cells monolayer in a T-25 cell culture flask. After 1 h of incubation at 37 °C, 5 mL of maintenance medium (EMEM supplemented with 2% FBS, 1% penicillin-streptomycin, 1% (200 mM) L-glutamine, 1% Kanamycin, and 3% Amphotericin B) was added. The flasks were incubated at 37 °C and examined daily for cytopathic effects. Two passages were performed successively on day 7 post-infection, and supernatants were harvested and clarified. Then, 200 µL aliquot of each passage was tested using phlebovirus generic RT-Nested-PCR as previously described [28].

### 2.5. Sequencing and Phylogenetic Analysis

Positive PCR products were purified using ExoSAP-IT™ Express PCR Product Clean-up (Applied Biosystem, California, Uinted States of America) and sequenced using the sanger technology. The sequencing reaction was performed with the Big Dye Terminator V3.1 Kit (Applied Biosystems, Waltham, MA, USA) with the same primers and cleaned up with Sephadex^®^ G-50 Superfine (SIGMA-ALDRICH, United States of America). The purified products of the cycle sequencing were analyzed on the ABI 3130 Genetic Analyzer (Applied Biosystems, Waltham, MA, USA). Sequences generated were aligned using MAFFT software [29], with homologous sequences recovered from the GenBank database. The phylogenetic tree was built by the Maximum Likelihood method in MEGA X software [30]. The robustness of the nodes was tested using 500 bootstrap replicates.

### 2.6. Sand Flies Identification

#### 2.6.1. Morphological Identification of Sand Flies

Sand fly species fauna was determined from 10% of the caught sand flies. Sand flies were treated in NaOH 10% and fixed in Marc-André solution according to the Abonnec protocol [31]. Each specimen was slide mounted in Marc-André solution and examined under a light microscope. Male genitalia and female spermathecae were used to identify specimens at the species level using morphological identification keys [31,32,33].

#### 2.6.2. Molecular Identification of Positive Pools

To confirm the sand fly species in the positive pools, the pellets of the positive pools were collected and used for DNA extraction with the DNA-RNA Extraction Mini Kit (Da An Gene Co., Ltd. of Sun Yat University, Guangzhou, China). PCR Reaction was conducted with 5µL of DNA added to 45 µL of Mix containing 1.25 U of Hot StarTaq Plus Master Mix (Qiagen, Hilden, Germany), 2× concentrated. Contains HotStarTaq Plus DNA Polymerase, PCR Buffer (with 1.5 mM Mg Cl 2), and 0.2 mM each dNTP, and 10 pmol of each universal barcoding primers LCO:5′-GGTCAACAAATCATAAAGATA TTGG-3, and HCO:5′-TAAACTTCAGGGTGA CCAAA AAATCA-3′, to amplifying 700 bp mtDNA genome of the cytochrome oxidase I region as previously described by [34], using the following thermal conditions: initial denaturing at 95 °C for 5 min, then 37 cycles of 94 °C for 30 s, 55 °C for 30 s and 72 °C for 90 s, followed by a 10 min extension at 72 °C. The PCR products were visualized on 2% agarose gels. The positive products were purified and sequenced in both directions using the same primers. The sequences were edited in BioEdit software [35], compared to those published sequences in GenBank.

## 3. Results

### 3.1. Virus Detection and Sand Flies Identification

A total of 3351 sand flies were captured in Kherrata (Bejaia) of whom 3051 were grouped into 62 pools (19 males and 43 females) and used for virus detection and 300 for morphological identification (Table 1). Phlebovirus was detected in two pools Ph9 and Ph33 (female and male, respectively) using RT-Nested-PCR, hence the overall prevalence of phlebovirus infection among the collected sand flies was 0.06% (2/3051) (Table 1). The infection rate = number of positive pools/number of sand flies ×100 [18].

### 3.2. Virus Isolation

Pools Ph9 and Ph33 showed clear cytopathic effects on passage 0 on days 4 and 5, respectively. Phlebovirus isolation was confirmed by RT-Nested-PCR as described before. The consensus sequences from both strains obtained from the first step of the RT-PCR have a size of 550 pb and both have been submitted to GenBank as PUNV strain Ph9 (access number ON524174) and Ph33 (access number ON524173).

### 3.3. Phylogenetic Analysis

Two new strains of PUNV have been detected in the phylogenetic analysis carried out in a short L-fragment (550 pb) (Figure 2). The maximum likelihood tree showed the relationship between ON524174 PUNV strain Ph9 and the PUNV strain isolated in Tunisia, it was supported by 76% bootstraps and with 3.3% and 0% of nucleotide and amino acid distances, respectively, while ON524173 PUNV strain Ph33 is related to the PUNV strain detected in Algeria and their relationship was supported by 61% bootstraps and with 3.5% and 0% of nucleotide and amino acid distances, respectively as described in (Figure 2 and Table 2).

### 3.4. Sand Fly Identification

The morphological identification showed the presence of three species in the study area, belonging to the subgenus *Larroussius. Ph. perniciosus* (98.67%) was the most present species as expected followed by *Ph. perfiliewi* (0.67%) and *Ph. longicuspis* (0.67%). *Ph. perniciosus* was identified in both positive pools using LCO1490/HCO2950 (Table 3 and Table 4).

## 4. Discussion

To date, only a little research regarding phleboviruses were carried out in North Africa and particularly in Algeria, even though sand flies can transmit a wide range of arthropod-borne viruses belonging to *Phenuiviridae*, *Reoviridae*, and *Rhabdoviridae* families compared to the parasitic diseases such as leishmaniasis [9]. Until now, only TOSV was isolated from Algerian sand flies [17]. However, it has been suggested that around the Mediterranean basin new phleboviruses are commonly found [36]. In our investigation, male and female Algerian sand flies were found infected with Punique virus and this was previously reported with other phleboviruses by several studies [18,37,38,39,40], this could be explained either by transovarial (vertical) or venereal (horizontal) virus transmission [41,42,43], suggesting that phleboviruses maintenance in nature is mainly guaranteed by sand flies. Humans and large vertebrates could amplify the virus but they are generally considered dead-end hosts because until now there is no evidence of their implication as reservoirs for these viruses, subsequently, they do not have an important role in the natural life cycle of the virus [44]. Phylogenetic analysis shows that the two strains isolated in our study (Ph9 and Ph33) form a strong cluster (100% bootstrap support) with the PUNV strains from Tunisia and Algeria, respectively (Figure 1). The ON524174 PUNV strain Ph9 was closely related to the sequence AB905362.1 PUNV strain from Tunisia with nucleotide and amino acid distances of 3.3 and 0%, respectively. While ON524173 PUNV strain Ph33 was closely related to the sequence MT250046.1 PUNV strain Pu0518.0.2_2018 from Algeria, detected in Blida, the nucleotides and amino acid distances were, respectively, 3.5% and 0%. In addition, ON524174 PUNV strain Ph9 and ON524173 PUNV strain Ph33 strains were detected and isolated from the same animal shelter they exhibited a genetic difference of 5% and 0.6% in amino acid distances suggesting a different source of the virus in the study area. The global infection rate of the sand flies by PUNV captured in our study area was 0.06% (2/3051 sand flies), less important compared with data reported in Tunisia at 0.13% [18] and remained quite similar to the infection rate with TOSV observed in Spain of 0.05% [45], Tunisia at 0.03% [13], and France at 0.1% [46]. Nevertheless, higher infection rates were reported for SFS-like virus and SFNV at 0.43%, and 0.25%, respectively [19,20], in Algeria. A lower infection rate with TOSV was noticed at 0.005% in Algeria [17]. All sand flies captured in our study belong to the subgenus *Larroussius*, *Ph. perniciosus* (98.67%), *Ph. perfiliewi* (0.67%), and *Ph. longicuspis* (0.67%). The dominant species was, as expected, *Ph. perniciosus* because it is the main species identified in the humid and sub-humid bioclimate stages [2,33], this species was confirmed in the positive pools by molecular technique. PUNV was first isolated from *Ph. perniciosus*, and detected in *Ph. longicuspis*, from the Utique region, a well-known focus of human visceral leishmaniasis, which is transmitted in Tunisia principally by *Ph. perniciosus* [18,47]. In addition, in our study, the two strains of PUNV were isolated from sand flies pools of *Ph. perniciosus* in an area where visceral leishmaniasis is endemic [25], this raises the question of the possible co-circulation of *Leishmania infantum* and PUNV in the same area. A co-circulation of *Leishmania infantum* and TOSV were previously observed in a focus on visceral leishmaniasis in central Tunisia [23]. So far, the impact of the PUNV on public health is unknown. Even though a co-circulation of PUNV and TOSV has been proven in Tunisia, there is not enough data to prove its pathogenicity for humans [48]. However, the fact that PUNV is a member of the SFNV-group (the majority of whose members are pathogenic to humans) and that it is detected in *Ph*. *perniciosus*, a proven vector of *Leishmania infantum* in Algeria [49], implies the need to conduct studies on the impact of this virus on public health. It is important to highlight that 12 years have passed between the discovery of the TOSV in *Ph. perniciosus* in 1971 [50] and the first evidence that it is a pathogen for humans [51].

## 5. Conclusions

Our results allowed the isolation for the first time in Algeria, of two different strains related to PUNV from sand flies belonging to the subgenus *Larroussius*. The infection was observed in both male and female sand flies. In the study area, the dominant species was *Ph. perniciosus*, a known vector of *Leishmania infantum* and TOSV, also, the DNA of this species was the most dominant in the positive pools. Currently, there is no effective vaccine or treatment against human pathogenic phleboviruses such as TOSV, the only solution that can protect against phleboviruses would be the application of collective and individual control measures against sand flies. Serological investigation of patients with fever of unknown origin, meningitis, and/or encephalitis is needed to determine the impact of this virus on public health, as well as entomological and serological investigations in domestic animals.

## Figures and Tables

**Figure 1 viruses-14-01796-f001:**
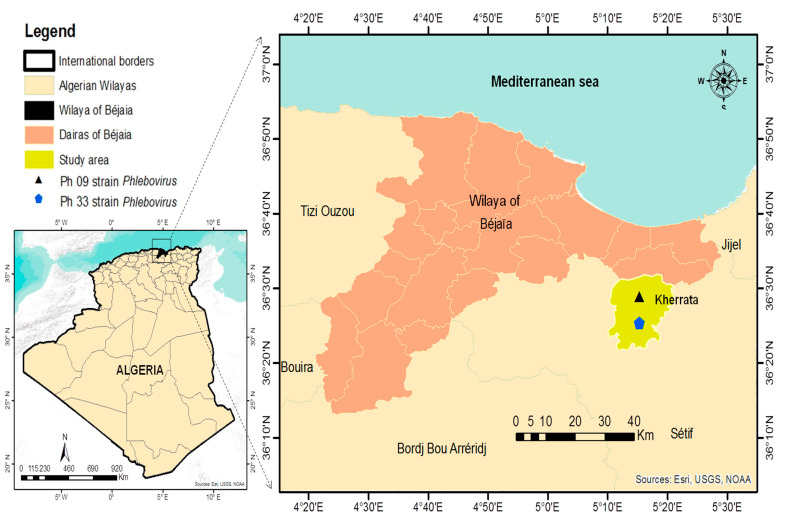
Geographical location of the study area.

**Figure 2 viruses-14-01796-f002:**
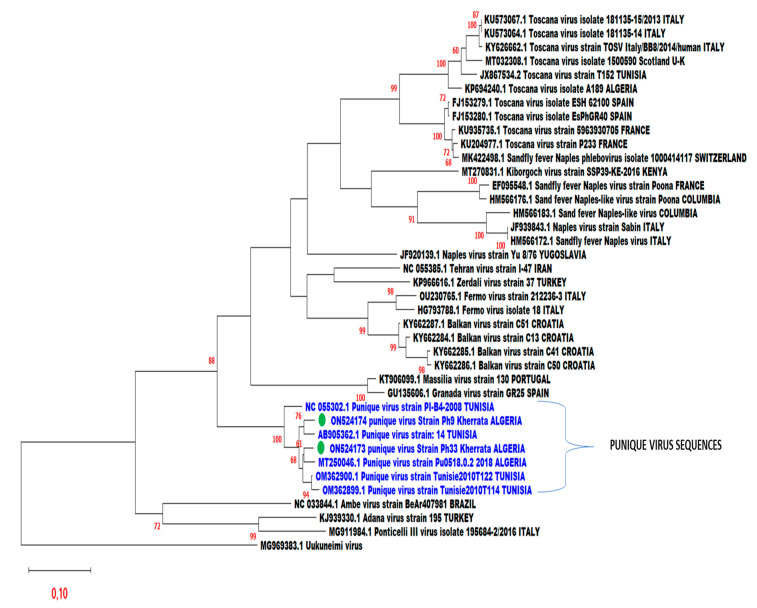
Phylogenetic analysis of the two strains isolated in the study area. Distances and groupings were determined by the pairwise-distance algorithm and Maximum Likelihood method within MEGAX, and the robustness of the groups was tested using 500 bootstrap pseudo-replicates.

**Table 1 viruses-14-01796-t001:** Number of sand flies used for conventional PCR and infection rate.

	Male	Female	Total
Number of sand flies	940	2111	3051
Number of pools	19	43	62
Number of positive pools	1(Ph33)	1(Ph9)	2
Infection rate (%)	0.1%	0.05%	0.06%

**Table 2 viruses-14-01796-t002:** Nucleotides and amino acid pairwise distance between Punique virus sequences.

N° Sequences	1	2	3	4	5	6	7
	Amino Acids Distances						
1. ON524174 PUNV strain Ph9 Kherrata Algeria		0.000	0.0061	0.0061	0.0381	0.0121	0.0121
2. AB905362.1 PUNV strain 14 Tunisia	0.0327		0.0061	0.0118	0.0381	0.0121	0.0236
3. ON524173 PUNV strain Ph33 Kherrata Algeria	0.0500	0.0412		0.000	0.0305	0.0061	0.0182
4. MT250046.1 PUNV strain Pu0518.0.2 2018 Algeria	0.0476	0.0515	0.0348		0.0305	0.0061	0.0287
5. OM362899.1 PUNV strain2010T114 Tunisia	0.0590	0.0561	0.0478	0.0505		0.0229	0.0229
6. OM362900.1 PUNV strain 2010T122	0.0412	0.0369	0.0328	0.0348	0.0128		0.0121
7. NC 055302.1 PUNV straiPI-B4-2008 Tunisia	0.0632	0.0770	0.0676	0.0763	0.0705	0.0542	
	Nucleotides distances						

**Table 3 viruses-14-01796-t003:** Species identified in the study area.

	Sex			
Species	Male	Female	(%)	Total
*Ph. perniciosus*	152	144	98.97%	296
*Ph. longicuspis*	0	2	0.67%	2
*Ph. perfiliewi*	1	1	0.67%	2
Total	153	147	100%	300

**Table 4 viruses-14-01796-t004:** Sand fly species in the positive pools after sequencing.

Positive Pools	Phlebovirus	Sand Flies Species
Ph 9	Punique virus	*Ph. perniciosus*
Ph 33	Punique virus	*Ph. perniciosus*

## Data Availability

Not applicable.

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
