# Peer review of "First Isolation of Punique Virus from Sand Flies Collected in Northern Algeria"

_viruses, 2022, doi:10.3390/v14081796_

Round 1

Reviewer 1 Report

In this manuscript two Punique virus were isolated from sandflies collected in Kherrate, Algeria. It's the second time this virus was detected in Algeria, suggesting its broad circulation in this country.

The content of the manuscript is rich. However, if the whole genome sequences of the isolates can be got and analysed, and if the photographs of cytopathic effects even Electron microscopy can be shown, it would be better.

Some minor suggestions.

1, Line 50, PUNV was used firstly in the text, it should be used as complete name.

2, Please rearrange the sentences in Line 62 and line 63. It seemed that "the" between "was" and "first" should be deleted, and the sentence "Followed..." is not completed.

3, Line 150, "MAFT" should be "MAFFT".

4, Line 182, How to define the infection rate should be explained, and the reference should be added.

Author Response

- Question1.  The content of the manuscript is rich. However, if the whole genome sequences of the isolates can be got and analysed, and if the photographs of cytopathic effects even Electron microscopy can be shown, it would be better.

- Answer 1: We thank the reviewer for this crucial point for improving the quality of the paper. However, actually we are unable to complete it because we don't have a light microscope picture of cytopatic effect in the laboratory at Pasteur Institute of Algieria, and the complete genome is not analyzed,  we are going to do it in the near future according to financial logistics and programs.

- Question 2. Line 50, PUNV was used firstly in the text, it should be used as complete name.

- Answer 2: As asked by the reviewer the PUNV is now puted in parenthesis and the complete name of the virus is montiened.

- Question 3. Please rearrange the sentences in Line 62 and line 63. It seemed that "the" between "was" and "first" should be deleted, and the sentence "Followed..." is not completed.

- Answer 3: Thanks to the reviewer for this pertinent remark. As proposed, the sentence is now corrected and rearranged as below (However, in Algeria, few data on SBPS are available, in 2006 Sicilian-like virus was detected  for the first time in Ph. ariasi and later in Ph. papatasi [19, 20].) 

- Question 4. Line 150, "MAFT" should be "MAFFT".

- Answer 4: As proposed by the reviewer, the word is now corrected in the text. 

- Question 5. Line 182, How to define the infection rate should be explained, and the reference should be added.

- Answer 5: We agree with the reviewer for this remark. As proposed, information about infection rate and reference are now addeded in the text as below ( The infection rate = Number of positive pools / Number of sand flies ×100 [18].)

Reviewer 2 Report

General comments:

The aim of the study was to investigate the circulation of phleboviruses among sand flies in a well-known endemic area of human and canine leishmaniasis to understand better the circulation of sand fly-born phleboviruses (SBPs), in Algeria. 

The results are clears and supported by the discussion. The conclusion are correct. References are appropriate.

Minor essential comments:

Tables and Figures should be improved:

Table 1:  Add the name of the positive pools in parenthesis (Ph33) and (Ph09), near the number 1, in the male and female pools columns respectively. 

Figure 2: In the phylogenetic tree show only the bootstrap values above 60.

Check the correct spelling of "sand fly" through the entire manuscript including the title.  

Author Response

- Question 1. Tables and Figures should be improved:

- Answer 1: We thank the reviewer for this remark to improve the quality of the paper. As proposed, figures and tables are now revised and improved.   

- Question 2. Table 1:  Add the name of the positive pools in parenthesis (Ph33) and (Ph09), near the number 1, in the male and female pools columns respectively. 

- Answer 2: As proposed by the reviewer, corrections are faithfully reported to the table 1.

- Question 3.  Figure 2: In the phylogenetic tree show only the bootstrap values above 60.

- Answer 3: We thank the reviewer for this excellent remark. The  phylogenetic tree is now corrected in the text.

- Question 4. Check the correct spelling of "sand fly" through the entire manuscript including the title.  

- Answer 4: As suggested by the reviewer, the correct spelling of "sand fly" is corrected through the title and in the text.
